# Unadversarial Examples: Designing Objects for Robust Vision

**Hadi Salman**[*]
hady@mit.edu
MIT[†]

**Andrew Ilyas**[*]
ailyas@mit.edu
MIT

**Logan Engstrom**[*]
engstrom@mit.edu
MIT

**Sai Vemprala**
saihv@microsoft.com
Microsoft Research

**Aleksander Mądry**
madry@mit.edu
MIT

**Ashish Kapoor**
akapoor@microsoft.com
Microsoft Research

## Abstract

We study a class of computer vision settings wherein one can modify the design of the objects being recognized. We develop a framework that leverages this capability—and deep networks' unusual sensitivity to input perturbations—to design "robust objects," i.e., objects that are explicitly optimized to be confidently classified. Our framework yields improved performance on standard benchmarks, a simulated robotics environment, and physical-world experiments.[1]

## 1 Introduction

Performing reliably on unseen or shifting data distributions is a difficult challenge for modern computer vision systems. For example, slight rotations and translations of images suffice to reduce the accuracy of state-of-the-art classifiers [ETT+19; ALG+19; KMF18]. Similarly, models that attain near human-level performance on benchmarks exhibit significantly degraded performance when faced with even mild image corruptions and transformations [HD19; KSH+19]. In fact, when an adversary is allowed to modify inputs directly, standard vision models can be manipulated into predicting arbitrary outputs (cf. *adversarial examples* [BCM+13; SZS+14]). While robustness interventions and additional training data can improve out-of-distribution behavior, they do not fully close the gap between model performance on standard heldout data and on corrupted/otherwise unfamiliar data [TDS+20; HBM+20]. The situation is worse still when test-time distribution is under- or mis-specified, which occurs commonly in practice.

How can we change this state of affairs? We propose a new approach to image recognition in the face of unforeseen corruptions or distribution shifts. This approach is rooted in a reconsideration of the problem setup itself. Specifically, we observe that in many situations, a system designer actually controls, to some extent, the inputs that are fed into that model. For example, a drone operator seeking to train a landing pad detector can modify the surface of the landing pad; and, a roboticist training a perception model to recognize a small set of custom objects can slightly alter the texture or design of these objects.

We find that such control over inputs can be leveraged to drastically improve our ability to tackle computer vision tasks. In particular, it allows us to turn the input-sensitivity of modern vision systems from a weakness into a strength. Instead of optimizing inputs to *mislead* models (e.g., as in adversarial examples), we can alter inputs to *reinforce* correct behavior, yielding what we refer to as "unadversarial examples." Indeed, we show that even a simple gradient-based algorithm can

---

[*]Equal contribution.

[†]Work partially completed while at Microsoft Research.

[1]Our code is available at https://github.com/microsoft/unadversarial.

35th Conference on Neural Information Processing Systems (NeurIPS 2021).

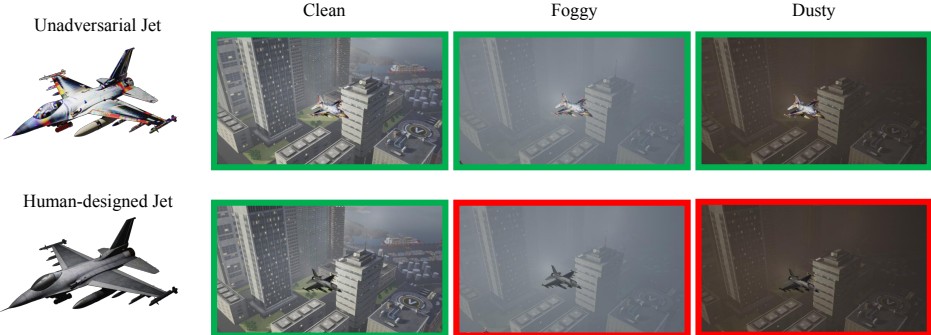

Figure 1: We demonstrate that optimizing objects (e.g., the pictured jet) for pre-trained neural networks can boost performance and robustness on computer vision tasks. Here, we show an example of classifying an unadversarial jet and a standard jet using a pretrained ImageNet model. The model correctly classifies the unadversarial jet even under bad weather conditions (e.g., foggy or dusty), whereas it fails to correctly classify the standard jet.

successfully construct unadversarial examples in a variety of vision settings and demonstrate that, by optimizing objects for vision systems (rather than vice-versa), we can significantly improve both in-distribution performance *and* robustness to unforeseen data shifts and corruptions.

We demonstrate the efficacy of our methods on both standard benchmarks (CIFAR, ImageNet) and robustness-based benchmarks (ImageNet-C, CIFAR-C) while also comparing them to a broad set of baselines (e.g., QR codes or heuristically designed patches). To further highlight the practicality of our framework, we (a) extend our methods to designing the texture of three-dimensional objects (rather than patches); (b) deploy unadversarial examples in a simulated drone setting; and (c) ensure that the performance improvement yielded by the objects we design actually transfer to the physical world.

## 2 Motivation and approach

While vision models tend to perform well on held-out data drawn from the same distribution as the training data, out-of-distribution inputs can severely degrade this performance. For example, models behave unreliably under distribution shifts induced by new data collection procedures [RRS+19; EIS+20; TE11], synthetic corruptions [HD19; KSH+19], spatial transformations [ETT+19; FF15], as well as under other types of shift.

Given a fixed type of distribution shift, a standard approach to increasing model robustness is to explicitly train on or regularize with data from the corresponding anticipated test distribution [KSH+19]. For example, Engstrom et al. [ETT+19] find that vision models trained on worst-case rotations and translations end up being fairly robust to rotation and translation-based distribution shifts. However, this approach is not without shortcomings—for example, Kang et al. [KSH+19] find that training CIFAR classification models that are robust to JPEG-compression in this manner requires a significant sacrifice in natural accuracy. Recent works make similar observations in the context of other distribution shift mechanisms like $\ell_p$ adversaries [TSE+19; SZC+18; RXY+19] or texture swapping [GRM+19].

These observations give rise to a more general question: given that performing reliably in the face of constrained, well-specified distribution shifts is already a difficult challenge, how can we attain robustness to broad, unforeseen distribution shifts?

### 2.1 Leveraging more controlled vision settings

Consider the vision tasks of detecting a landing pad from a drone, or classifying manufacturing components from a factory robot. In both these tasks, reliable in-distribution performance is a necessity; still, a number of possible distribution shifts may occur at deployment time. For example, the drone might approach the landing pad at an atypical angle, or have a view obstructed by snow,

smoke, or rain. Similarly, the factory robot may encounter objects in unfamiliar poses, or could be equipped with only a low-quality/noisy camera.

At first glance, dealing with these issues seems to require tackling the difficult problem of general distribution shift robustness discussed earlier in this section. However, there is in fact a critical distinction between the scenarios considered above and vision tasks in their full generality. In particular, in these scenarios and many others, the system designer has control over the physical objects that the model operates on. For instance, the designer of the drone's landing algorithm could paint the landing pad bright yellow. A machine learning model trained to detect this custom landing pad might then be more effective than a model trained to detect a standard grey pad, especially in low-visibility conditions. Still, the particular choice to paint the landing pad yellow is rather ad hoc, and likely rooted in the way *humans* recognize objects. Meanwhile, an abundance of prior work (e.g., [JBZ+19; GRM+19; JLT18; IST+19]) demonstrates that humans and machine learning models tend to use different sets of features to make their decisions. This suggests that rather than relying on human priors, we should instead be asking: *how can we build objects that are easily detectable by machine learning models?*

## 2.2 Unadversarial examples

The task of making inputs *less* recognizable by computer vision systems has been a focus of research in *adversarial examples*. Adversarial examples are small, carefully constructed perturbations to natural images that can induce arbitrary (mis)behavior from machine learning models [BCM+13; SZS+14]. These perturbations are typically constructed as the result of an optimization problem that maximizes the loss of a machine learning model with respect to the input, i.e., by solving the optimization problem

$$\delta_{adv} = \arg \max_{\delta \in \Delta} L(f_\theta(x + \delta), y), \tag{1}$$

where $f_\theta$ is a parameterized model (e.g., a neural network with weights $\theta$); $x$ is a natural input; $y$ is the corresponding correct label; $L$ is the loss function used to train $\theta$ (e.g., cross-entropy loss) and $\Delta$ is a class of permissible perturbations (e.g., norm-bounded perturbations: $\Delta = \{\delta : \|\delta\|_p \le \epsilon\}$ for some small $\epsilon > 0$). Adversarial perturbations are typically crafted via projected gradient descent (PGD) [Nes03] in input space, a standard iterative first-order optimization method—prior work in adversarial examples has shown that even a few iterations of PGD suffice to completely change the prediction of many state-of-the-art machine learning systems [MMS+18].

**From adversarial examples to unadversarial objects.** The goal of this work is to modify the design of objects so that they are more easily recognizable by computer vision systems. If we could specify every pixel of every image that a model encounters at test time, we could draw on the effectiveness of adversarial examples, and construct image perturbations (using PGD) that *minimize* the loss of the system, e.g.,

$$\delta_{unadv} = \arg \min_{\delta \in \Delta} L(\theta; x + \delta, y). \tag{2}$$

In our setting of interest, however, having such fine-grained access to the test inputs is unrealistic (presumably, if we had precise control over every pixel in the input, we could just directly encode the ground-truth label directly in it). Instead, we have *limited* control over some physical objects; these objects are in turn captured within images, affected by many signals that are out of our control, such as camera artifacts, weather effects, or background scenery.

It turns out that we can still draw on techniques from adversarial examples research in this limited-control setting. Specifically, a recent line of work [KGB17; SBB+16; EEF+18a; AEI+18] concerns itself with constructing *robust adversarial examples* [AEI+18], i.e., physically realizable objects that act as adversarial examples when introduced into a scene in any one of a variety of ways. For example, Sharif et al. [SBB+16] design glasses frames that cause facial recognition models to misclassify faces, Athalye et al. [AEI+18] design custom-textured 3D models that are misclassified by state-of-the-art ImageNet classifiers from many angles and viewpoints, and [BMR+18] design adversarial patches: stickers that can be placed anywhere on objects causing them to be misclassified. In this paper, we leverage the techniques developed in the above line of work to construct robust un-adversarial objects—physically realizable *objects* optimized to minimize (rather than maximize) the loss of a target classifier. In the next section, we will more concretely discuss our methods for generating unadversarial objects, then outline our evaluation setup.

Unadv patch

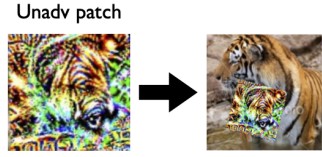

(a) An example unadversarial patch designed for the "tiger" class.

Unadv texture

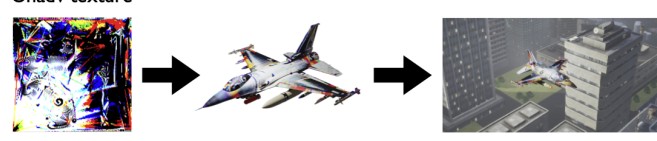

(b) An example unadversarial texture designed for a jet 3D mesh (class "warplane") and applied to rendered city backgrounds.

Figure 2: Examples of the two considered methods for constructing unadversarial objects.

## 2.3 Constructing unadversarial objects

In the previous section, we identified a class of scenarios where a system designer can, to some extent, control the objects that a machine learning system model operates on. In these settings, we motivated so-called *unadversarial examples* as a potential way to boost models' overall performance and robustness to distribution shifts. In this section, we present and illustrate two concrete algorithms for constructing unadversarial examples: unadversarial patches and unadversarial textures. In the former, we design a sticker or "patch" [BMR+18] that can be placed on the object; in the latter, we design the 3D texture of the object (in a similar manner to the texture-based adversarial examples of Athalye et al. [AEI+18]). Example results from both techniques are shown in Figure 2. For simplicity, we will assume that the task being performed is image classification, but the techniques are directly applicable to other tasks as well. In all cases, we require access to a pre-trained model for the dataset of interest.

**Unadversarial patches (cf. Algorithm 1 in Appendix A).** To train unadversarial patches (cf. Figure 2a), in addition to the pre-trained model, we require sample access to image-label pairs from the dataset of interest. At each iteration, we sample an image-label pair $(x, y)$ from a training set, and place the patch corresponding to class $y$ onto the image with random orientation and position[2]. Since placing the patch is an affine transformation, after each iteration we can compute the gradient of the model's loss with respect to the pixels in the patch, and take a negative gradient step on the patch parameters. The algorithm terminates when the model's loss on sticker-boosted images plateaus, or after a fixed number of iterations.

**Unadversarial textures (cf. Algorithm 2 in Appendix A).** To train unadversarial *textures* (cf. Figure 2b), we do not require sample access to the dataset, but instead a set of 3D meshes for each class of objects that we would like to augment, as well as a set of background images that we can use to simulate sampling a scene (these can be images from the dataset of interest, solid-color backgrounds, random patterns, etc.).

For each 3D mesh, our goal is to optimize a 2D texture which improves classifier performance when mapped onto the mesh. At each iteration, we sample a mesh and a random background; we then use a 3D renderer (Mitsuba [NVZ+19]) to map the corresponding texture onto the mesh. We overlay the rendering onto a random background image, and then feed the resulting composed image into the pre-trained classifier, with the label being that of the sampled 3D mesh. Since rendering is typically non-differentiable, we use a linear approximation of the rendering process (cf. Athalye et al. [AEI+18]) in order to compute (this time approximate) gradients of the model's loss with respect to the utilized texture. From there, we apply the same SGD algorithm as we did for the patch case.

## 3 Experimental evaluation

In order to determine the effectiveness of our proposed framework, we evaluate against a suite of computer vision tasks. Below, we first provide some detail on the precise access model and baselines that will be considered. We then briefly outline the experimental setup of each task, and show that

---

[2]We allow the patch to be placed anywhere as a matter of convenience: ideally we would only be applying the patch onto the main object itself, but this would require bounding box data that we do not have for most classification datasets.

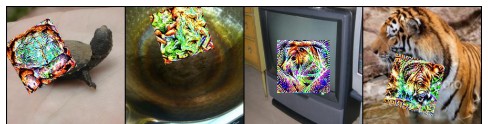 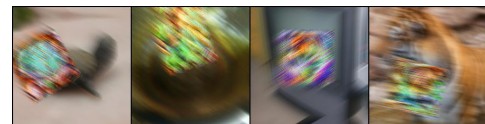

Figure 3: Clean (left) and corresponding corrupted (right) ImageNet images augmented with an unadversarial patch—we use such images to evaluate the efficacy of unadversarial patches in Section 3.2.

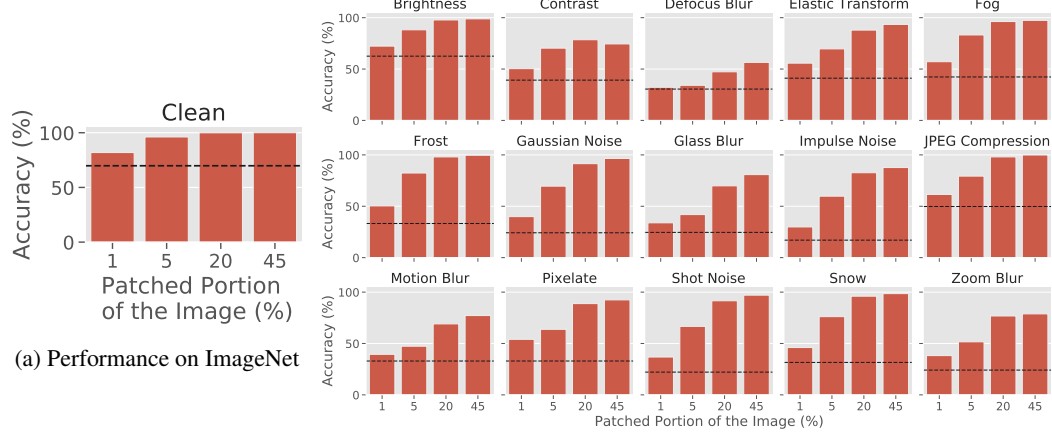

(a) Performance on ImageNet

(b) Performance on synthetically corrupted data (ImageNet-C)

Figure 4: Accuracy on (a) clean ImageNet images and (b) synthetically corrupted ImageNet-C images as a function of patch size (given as a percentage of image area). In (b), each bar denotes the average accuracy over the five severities in ImageNet-C, and the horizontal dashed lines report the accuracy on the original (non-patched) datasets. Unadversarial patches consistently boost performance for both clean and corrupted images, with accuracy monotonically increasing with patch size. The patches were trained without any corruptions or non-standard data augmentation in-the-loop (we train with the same augmentations that the pre-trained model itself was trained with).

unadversarial objects consistently improve the performance and robustness of the vision systems tested. For a more detailed account of each experimental setup, see Appendix C.

## 3.1 Access model and baselines

In many of the settings discussed thus far, a system designer can alter the objects being recognized but is *not allowed* to alter the the classifier itself. That is, we are optimizing unadversarial objects for a fixed (pre-trained) model. For instance, a road engineer may wish to design road signs that are easier to recognize for autonomous vehicles, without being able to train or alter the machine learning models that operate the vehicles. Similarly, a roboticist might want to design a landing pad that works better for a commercial (pre-trained) drone. We will refer to this setting as the *fixed-model* setting. On the other hand, sometimes the same entity is able to train both the model and the transformations (the discussed factory robot example may fall into this category, for example). In this "free-model" setting, one may be able to boost performance by *co-designing* the machine learning model and the objects of interest.

In this work, we will focus on designing unadversarial examples in the fixed-model setting, for the sake of both simplicity and applicability. In particular, any valid algorithm under a fixed-model assumption is also a valid algorithm under the co-design assumption (but the converse is not true). This leaves the task of leveraging even more control in joint optimization settings as a potential avenue for future work.

**Baselines.** Since we consider the fixed-model setting throughout our paper, the only truly comparable baselines are those which do not alter the model being trained. Nonetheless, in order to fully contextualize our results, we will also consider a few baselines that fall outside of our intended access model (e.g., QR codes). A notable disadvantage of these baselines (that we do not explicitly

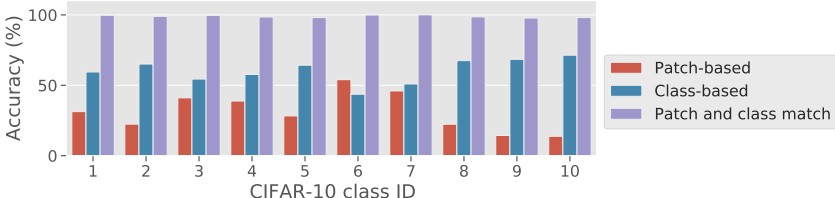

Figure 5: The accuracy of a pretrained ResNet-50 on boosted CIFAR-10 when the image and the unadversarial patch used to boost the image have: (1) same class, (2) conflicting classes. For the conflicting classes setting, we report the accuracies based on the class of the patch (red bars), and those based on the class of the image (blue bars). When there is a conflicting signal between the patch and the image, the model relies more on the image.

demonstrate below) is that if the "unadversarial signal" (e.g., the QR code) is occluded or removed, the entire system fails; this is in contrast to the fixed-model setting, where the pre-trained model is able to recognize objects without any unadversarial signals and is instead "boosted" by their presence.

## 3.2   Clean data and synthetic corruptions

We first test whether unadversarial examples improve the performance of image classifiers on benchmark datasets. Using the algorithm described in Section 2.3, we construct unadversarial patches of varying size for pre-trained ResNet-50 classifiers on the CIFAR [Kri09] and ImageNet [RDS+15] datasets. For evaluation, we add these patches at random positions, scales, and orientations to validation set images (see Appendix C for the exact protocol). As shown in Figure 4a, the pre-trained ImageNet classifier is consistently more accurate on the augmented ImageNet images. For example, an unadversarial patch 20 times smaller than ImageNet images boosts accuracy by 26.3% (analogous results for CIFAR are given in Appendix D).

**Robustness to synthetic corruptions.** Next, we use the CIFAR-C and ImageNet-C datasets [HD19] (consisting of the CIFAR and ImageNet validation sets corrupted in 15 systematic ways) to see whether the addition of unadversarial patches to images confers any corruption robustness.

We use the same patches and evaluation protocol that we used when looking at clean data (to ensure a fair evaluation, we apply corruptions to boosted images only *after* the unadversarial patches have been applied). As a consequence, at test time neither model nor patch has been exposed to any image corruptions beyond standard data augmentation. As a result, this experiment tests the ability for unadversarially boosted images to withstand completely unforeseen corruptions; we also avoid any potential biases from training on (and thus "overfitting" to [KSH+19]) a specific type of corruption. The results (cf. Figure 4b for ImageNet and Appendix D for CIFAR) indicate that unadversarial patches do improve performance across corruption types; for example, applying an unadversarial patch 5% the size of a standard ImageNet image boosts accuracy by an average of 31.7% points across corruptions [3].

**The model does not ignore the image in the presence of unadversarial patches.**

Recall from our discussion of the fixed-model setting in Section 3.1 that an advantage of designing unadversarial objects without changing the model is that the model still works in the absence of the unadversarial signal. We now briefly explore the case where the model is exposed to an unadversarial signal for the *wrong class*. Ideally, we would want the patch to only assist/boost the signal from the original image—in particular, we do not want the patch to make the model totally ignore the contents of the image itself. Thus, in cases where the signal from the image and the patch conflict, we would like the classifier to predict according to the features present in the image more frequently than the class encoded in the unadversarial patch.

---

[3]Since the original corruption benchmarks proposed by [HD19] are only available as pre-computed JPEGs (for which we cannot apply a patch pre-corruption) or CPU-based Python image operations (which were prohibitively slow), we re-implemented all 15 corruptions as batched GPU operations; we verified that model accuracies on our corruptions mirrored the original CPU counterparts (i.e., within 1% accuracy). For more details about our reimplementation, see our code release.

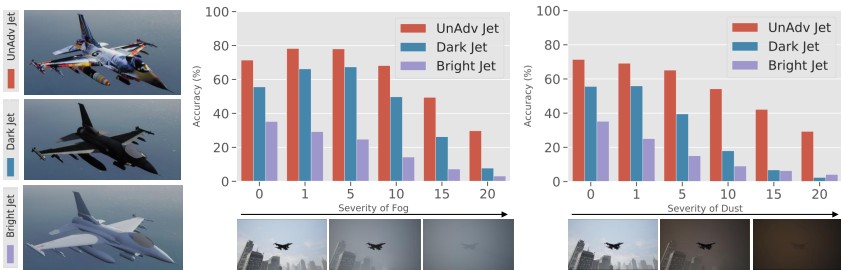

Figure 6: The jet unadversarial example task. We show example conditions under which we evaluate the objects, along with aggregate statistics for how well an ImageNet classifier classifies the objects in different conditions. We find that the classifiers perform consistently better on the unadversarial jet texture over the standard jet texture in both standard and distributionally shifted conditions. We also give a baseline of a white jet with a lighter texture because of the poorly visibility inherent in the simulator; we find it performed worse than even the standard jet.

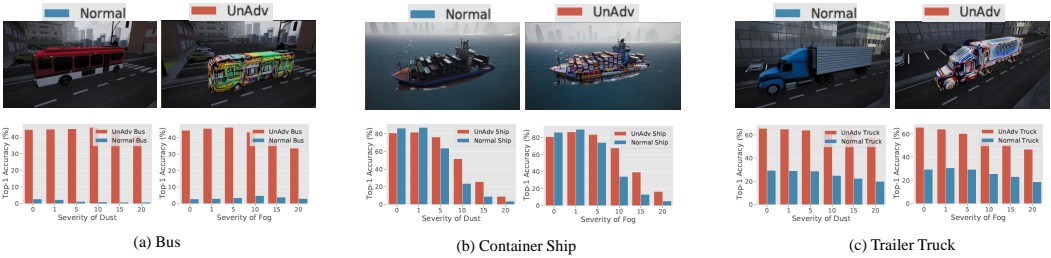

(a) Bus        (b) Container Ship        (c) Trailer Truck

Figure 7: Additional examples reporting aggregate statistics for how well an ImageNet classifier classifies various objects in different conditions. Again, we find that the classifiers perform consistently better on the unadversarial objects texture over the standard objects.

Indeed, this turns out to be the case. Figure 5 shows that on CIFAR-10, when the signal from an unadversarial patch and the image itself conflict, the model predicts according to the patch only 31.2% of the time on average, and according to the image 60.3% of the time (the accuracy of the model when the patch and image agree is 98.93%.)

At first glance, this result may seem to be at odds with the near-perfect effectiveness of *adversarial* patches [BMR+18]. However, the phenomenon we observe here can be tied to the subtle difference between the way we train our unadversarial patches and the way one trains targeted adversarial patches. In the former, we overlay each patch exclusively onto images from its respective class— thus, unadversarial patches are never optimized to be effective when overlaid on a different class. In the latter, however, adversarial patches are optimized to maximize confidence in a particular class on all possible backgrounds, making the patch dominant even when overlaid on an image from a different class.

**Baselines.** We also compare our results to a variety of natural baselines; the most relevant of these is the "best loss image patch," where we use the minimum-loss training image in place of a patch. We compare with this baseline to ensure that our method is doing something beyond this naive way to add signal to an image. The results are shown in Appendix D, along with comparisons to less sophisticated baselines, such as QR Codes and predefined random Gaussian noise patches.

### 3.3 Classification in 3D simulation

We now test unadversarial examples in a more practical setting: recognizing 3D objects in a high-fidelity simulator. We collect meshes corresponding to four ImageNet classes: "warplane," "minibus," "container ship," and "trailer truck," from `sketchfab.com`. We generate a texture for each object using the unadversarial texture algorithm of Section 2.3, using the ImageNet validation set as the set of backgrounds for the algorithm, and a pre-trained ResNet-50 as the classifier.

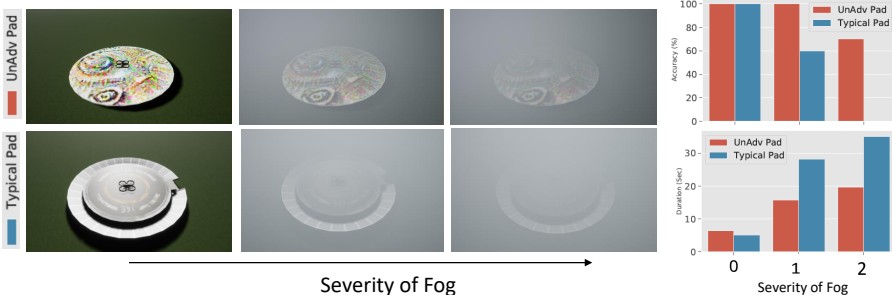

Figure 8: Drone landing task. On the left we show the unadversarial versus standard landing pads. On the right we show the results for the task when both the standard and unadversarial landing pads are used. We find that the drone consistently takes less time to land, and has a higher chance of landing correctly, when detecting the unadversarial landing pad.

To evaluate the resulting textures, we import each mesh into Microsoft AirSim, a high-fidelity three-dimensional simulator; we then test pre-trained ImageNet models' ability to recognize each object with and without the unadversarial texture applied in a variety of surroundings. We also test each texture's robustness to more realistic weather corruptions (snow and fog) built directly into the simulator (rather than applied as a post-processing step). We provide further detail on AirSim and our usage of it in Appendix B. Examples of the images used to evaluate the unadversarial textures, as well as our main results for one of the meshes are shown in Figure 6. We find that in both standard and adverse weather conditions, the model consistently performs better on the unadversarial texture than on the original. We present similar results for the other three meshes in Figure 7.

### 3.4 Localization for (simulated) drone landing

We then assess whether unadversarial examples can help outside of the classification setting. Again using AirSim, we set up a drone landing task with a perception module that receives as input an axis-aligned aerial image of a landing pad, and is tasked with outputing an estimate of the camera's $(x, y)$-position relative to the pad. While this task is quite basic, we are particularly interested in studying performance in the presence of heavy (simulated) weather-based corruptions. The drone is equipped with a pretrained regression model that localizes the landing pad (described in detail in Appendix B). We optimize an unadversarial texture for the surface of the landing pad to best help the drone's regression model in localization. Figure 8 shows an example of the landing pad localization task, along with the performance of the unadversarial landing pad compared to the standard pad. The drone landing on the unadversarial pad consistently lands both more reliably.

### 3.5 Physical-world unadversarial examples

Finally, we move out of simulation and test whether the unadversarial patches that we generate can survive naturally-arising distribution shift from effects such as real lighting, camera artifacts, and printing imperfections. We use four household objects (a toy racecar, miniature plane, coffeepot, and eggnog container), and print out (on a standard InkJet printer) the adversarial patch corresponding to the label of each object. We take pictures of the toy with and without the patch taped on using an ordinary cellphone camera, and count the number of poses for which the toy is correctly classified by a pre-trained ImageNet classifier. Our results are in Table 9a, and examples of patches are in Figure 9b. Classifying both patched and unpatched images over a diverse set of poses, we find that the adversarial patches consistently improve performance even at uncommon object orientations.

## 4 Related work

Here, we first highlight (and differentiate from) previous works using reference markers to improve recognition and localization. We will then discuss related work in adversarial robustness.

**Improving computer vision with fiducial markers.** In past research, vision-based precision landing was initially attempted with classical computer vision based tracking of reference designs

| Class | No Patch | Patch |
|---|---|---|
| "racer" | 22% | 83% |
| "eggnog" | 22% | 44% |
| "coffee pot" | 39% | 56% |
| "warplane" | 67% | 83% |

(a) Accuracy of pre-trained ResNet-18 on photographs of real world objects with and without patches.

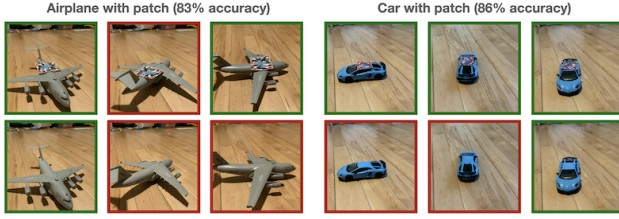

(b) Examples photos of the "warplane" and "racer" physical objects taken with (top) and without (bottom) an unadversarial patch.

Figure 9: Physical-world experiments. We take pictures of objects at diverse orientations while varying the presence of a patch on the object. Note that we don't do any additional data augmentation on the patches, which are the same used in our previous ImageNet benchmark experiment.

[SMS03; LSP09; YSS+14; FZS+17], or on arrangements of fiducial markers [LZT+19] (a fiducial marker is a fixed pattern or object that is placed in a scene as a reference point for location or measure). Several types of visual fiducial markers were also proposed for robust tracking and pose estimation applications through classical vision [Fia05; GMM+14; Ols11; RMM19]. While fiducial markers such as AprilTags [Ols11] were commonly used in robotics, they suffer from limitations such as orientation uncertainty, accuracy falloff depending on viewing angles, and short detection ranges [AAB+19; WO16], which motivated research into using convolutional neural networks for landing pad detection and pose estimation [NAK+18; TNN+19]. Our work attempts to unify these perspectives by leveraging the expressivity of neural network-based systems to design robust unadversarial examples. Another key difference between fiducial markers and unadversarial objects is that the former require vision systems to be aware of their presence; in contrast, the latter are designed on top of pre-trained systems. This means that we do not require any further non-standard model training, and that we do not depend as heavily on the unadversarial example being visible.

**Adversarial robustness.** Our design of unadversarial examples is motivated by the success of adversarial examples, i.e., minute perturbations to the inputs of machine learning models that can induce nearly arbitrary behavior. Adversarial examples were shown to be effective in seriously hampering machine vision related tasks such as classification [SZS+14; EEF+18a; AEI+18], object detection/segmentation [EEF+18b; AMT18; LK19; XWZ+17] and visual question-answering (VQA) [XCL+18]. Furthermore, prior work has shown that adversarial examples can be constructed even without direct access to the vision system being manipulated [CZS+17; PMG16; PMG+17; IEA+18; IEM18] Synthesized physical adversarial examples were shown to be effective in fooling person detection [TVG19], sign detection for autonomous driving [SBM+18; BHG+19]. Robotic platforms such as manipulators were also shown to be sensitive to vision based adversarial examples [MDB+17], and to specifically designed adversarial objects [WTL+19]. Additionally, recent work [EGS19; NHD+19] shows that one can "reprogram" neural networks using adversarial examples, e.g., one can construct a patch that causes a CIFAR-10 classifier to operate as an MNIST classifier.

## 5 Discussion and conclusions

In this work, we demonstrated that it is possible to design object alternations that boost the corresponding classifiers' performance, even under strong and *un*foreseen distribution shift. Indeed, such resulting unadversarial objects are robust to a broad range of data shifts and corruptions, even when these were never seen in training. We view our results as a promising route towards increasing out-of-distribution robustness of computer vision models.

**Limitations and future work.** One limitation of our method is that it requires differentiability with respect to properties of the input object of interest, or a close proxy (e.g. a differentiable simulator) that can mimic the object and the environment in which the object operates.

Furthermore, as discussed at length in Section 3.1, in this work we only consider the case where the system designer cannot alter the model being trained. While this carries some inherent advantages (e.g., increased applicability), future work should investigate whether attaining even more accuracy and robustness is possible by jointly optimizing models and unadversarial objects.

Finally, we have shown unadversarial examples to be effective only in classification and regression settings. However, the fact that unadversarial examples and adversarial examples share the same underlying generation technique is evidence that unadversarial examples could apply to any system that is vulnerable to adversarial examples. Thus, an avenue of future work could be to apply these examples to detection, segmentation, style transfer, and other domains.

**Broader impact.** On one hand, as with any improvement to the robustness and reliability of computer vision systems, unadversarial examples also carry the potential for misuse (e.g., as part of overreaching surveillance technology). On the other hand, unadversarial examples give end users of machine learning systems—who typically have little to no power over the behaviour of the ML systems they interact with—the power to improve the performance and robustness of these systems in their respective use cases. For example, in the future someone living in a remote area where roads or traffic signs may look "atypical" can slightly modify their signs to be better recognized by autonomous vehicles.

## Acknowledgements

We are grateful to Ian Engstrom for helping take the photographs for the physical-world experiments.

Work supported in part by the NSF grants CCF-1553428 and CNS-1815221, Open Philanthropy, and the Microsoft Corporation. This material is based upon work supported by the Defense Advanced Research Projects Agency (DARPA) under Contract No. HR001120C0015.

Research was sponsored by the United States Air Force Research Laboratory and the United States Air Force Artificial Intelligence Accelerator and was accomplished under Cooperative Agreement Number FA8750-19-2-1000. The views and conclusions contained in this document are those of the authors and should not be interpreted as representing the official policies, either expressed or implied, of the United States Air Force or the U.S. Government. The U.S. Government is authorized to reproduce and distribute reprints for Government purposes notwithstanding any copyright notation herein.

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
