# A   Pseudocode for Unadversarial Example Generation

---

**Algorithm 1:** Unadversarial patch generation

---

**Input:** Pre-trained classifier with parameters $w$, loss function $\ell_w(x, y)$, dataset $\mathcal{D}$
**Input:** Image size $m$, patch size $n$, target class $C_{targ}$, patch learning rate $\eta$
**Result:** An unadversarial patch for the class $C_{targ}$
Randomly initialize a patch $\theta \in \mathbb{R}^{n \times n \times 3}$;
**for** $K$ *iterations* **do**
    Sample batch of image-label pairs $(x, y) \sim \mathcal{D}$;
    **if** $y = C_{targ}$ **then**
        $\theta_{padded} \leftarrow$ Zero-pad $\theta$ to size $m \times m$;
        $mask \leftarrow \text{int}(\theta_{padded} > 0)$ ;          // 0/1 mask signalling patch location
        $T \leftarrow$ RandomAffineTransform() ;        // random rotation, scaling, and
        translation
        $x_{unadv} \leftarrow x \cdot (1 - T(mask)) + T(\theta_{padded}) \cdot T(mask)$ ;    // apply patch using
        mask
        $\theta \leftarrow \theta - \eta \cdot \text{sign}\left(\nabla_\theta \ell_w(x_{unadv}, y)\right)$ ;         // gradient descent step on $\theta$
    **end**
**end**
**return** $\theta$

---

**Algorithm 2:** Unadversarial texture generation

---

**Input:** Pre-trained classifier with parameters $w$, loss function $\ell_w(x, y)$, set of background
        images $\mathcal{D}_b$
**Input:** Texture size $n$, target 3D mesh $M_{targ}$, texture learning rate $\eta$
**Result:** An unadversarial texture for the mesh $M_{targ}$
Randomly initialize a texture $\theta \in \mathbb{R}^{n \times n \times 3}$;
Init a texture $t_{uv} \in \mathbb{R}^{n \times n \times 3}$ with $t_{uv}[i, j, 1] = i, t_{uv}[i, j, 2] = j, t_{uv}[i, j, 3] = 0$ ; // $t_{uv}$ is a
 UV map
**for** $K$ *iterations* **do**
    Sample background $x_{bg} \sim \mathcal{D}$;
    Sample a random 3D configuration (position and orientation) $C_{3D}$;
    $x_{rend} \leftarrow$ render $M_{targ}$ in configuration $C_{3D}$ with texture $\theta$ and background $x_{bg}$;
    $x_{uv} \leftarrow$ render $M_{targ}$ in configuration $C_{3D}$ with texture $t_{uv}$ and clear background;
    $x_{drend} \leftarrow$ linear (differentiable) approximation to $x_{rend}$, i.e.,

$$x_{drend}[i, j] = \begin{cases} x_{bg}[i, j] & \text{if } x_{uv}[i, j] \text{ is blank} \\ \theta[x_{uv}[i, j]] & \text{if } x_{uv}[i, j] \text{ is not blank} \end{cases}$$

    $x_{unadv} \leftarrow x_{drend} - \text{detach}(x_{drend}) + x_{rend}$ ;         // so $x_{unadv} = x_{rend}$ but
    $\nabla_\theta x_{unadv} = \nabla_\theta x_{drend}$
    $\theta \leftarrow \theta - \eta \cdot \text{sign}\left(\nabla_\theta \ell_w(x_{unadv}, y)\right)$ ;         // gradient descent step on $\theta$
**end**
**return** $\theta$

---

# B  3D Simulation Details

## B.1  Overview of AirSim

We conduct our simulation experiments using the high fidelity simulator, Microsoft AirSim. AirSim acts as a plugin to Unreal Engine, which is a AAA videogame engine providing access to high fidelity graphics features such as high resolution textures, realistic lighting, soft shadows etc. making it a good choice for rendering for computer vision applications. AirSim internally provides physics models for a quadrotor vehicle, which we leverage for performing autonomous drone landing. As a plugin, AirSim can be paired with any Unreal Engine environmnent to simulate autonomous vehicles that can be programmed with an API both in terms of planning/control as well as obtaining camera images. AirSim also allows for controlling environmental features such as time of day, dynamically adding/removing objects, changing object textures and so on.

## B.2  3D Boosters Classification Experiment

**Format of 3D models** To evaluate the performance of pretrained ImageNet classifiers at detecting 3D unadversarial/boosted objects (e.g. the jet shown in the main paper) among realistic settings, we set up an experiment using AirSim for image classification of common classes (warplane, car, truck, ship, etc). We pick the class of 'warplane' as our object class of interest download publicly available 3D meshes for this class from `www.sketchfab.com`. Using the open source 3D modeling software Mitsuba, we modify the object texture to match the boosted texture for the corresponding class, and then export these meshes into the GLTF format for ingestion into Unreal Engine/AirSim. This allows us to import the boosted objects into the AirSim framework, and spawn them as objects in any of the environments being created.

**Environment screenshots and description** Within AirSim, in the interest of generating realistic imagery, we simulate a city environment (Figure 10a). For this experiment, we use the Computer-Vision mode of AirSim, which does not simulate a vehicle but rather, gives the user control of a free moving camera, allowing us to generate data at ease from various locations and varying camera and world parameters.

**Sampling and evaluation** Once the 3D objects (unadversarial or normal) are present in AirSim's simulated world, the next step is to evaluate the classification of these objects from different camera angles, weather conditions etc. Given the location of a candidate object (which we randomize and average over five locations), we sample a grid ($10 \times 10 \times 10$) of camera positions in 3D around the object. For each of these positions, we move AirSim's main camera and orient it towards the object, resulting in images from various viewpoints. At runtime, each of these images are immediately processed by a pretrained ResNet-18 ImageNet classifier, which reports the top 5 class predictions. We average the accuracies across the five different locations in the scene and the 1000 grid points around the object at each location.

Along with this variation in camera angles and thereby, object pose in the frame; we also evaluate the performance of of the various 3D objects given environmental perturbations. We achieve this through the AirSim's weather conditions feature, using which we simulate weather conditions such as dust and fog dynamically with varying levels of severity of these conditions. We will open-source binaries for the AirSim code and environments that we use which will allow people to replicate our results, and investigate more scenarios of interest.

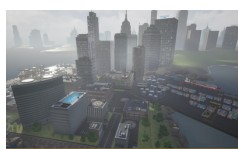 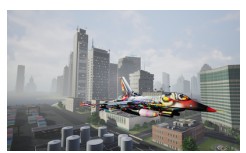 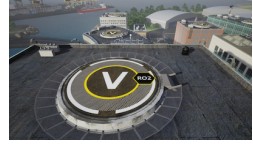 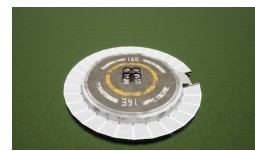

| (a) City environment in AirSim used for detection experiment | (b) Boosted 'jet' model in the City environment. | (c) Sample landing pads atop buildings in the City environment. | (d) Drone in test environment used for the landing experiments. |

## B.3 Drone Landing Experiment

In this experiment, we evaluate how unadversarial/boosted objects can help robustify perception-action loops that are driven by vision-based pose estimation. Perception-action loops are at the heart of many robotics tasks, and accurate perception is imperative for safe, efficient navigation of robots. We choose the scenario of autonomous drone landing as our experiment, and simulate it within AirSim.

For this experiment, we create assets of landing pads that are similar to helipads on top of buildings in the city environment(Figure 10c). We also use a test environment with a single landing pad located on a patch of grass. An example of such a landing pad can be seen in Figure 10d. We use AirSim to simulate a quadrotor drone in these worlds, which can be programmatically controlled using a Python API. AirSim allows us to equip a downward facing, gimballed camera on this drone in order to obtain RGB images, which are then processed by our landing pad pose estimation (regression) model. Given an RGB image, the regression model outputs a 6 degree of freedom pose for the landing pad. We use/optimize only the first two enteries of this output corresponding to the relative x and y location of the landing pad w.r.t the drone.

We formulate the drone landing experiment as a visual servoing task: a perception action loop that involves estimating the relative location of the pad from the image frame captured by the downward facing camera of the drone, and sending an appropriate velocity command in order to align the camera center with that of the pad. We achieve these through the following steps:

**Data Collection.** We use AirSim's inbuilt data collection API for this step. Given the location of the pad in the world, we sample various feasible locations for the drone in an imaginary cone whose vertex aligns with the center of the landing pad. We then spawn the drone in these randomly sampled positions, and obtain the RGB and segmentation views of the pad as generated by AirSim, along with the relative ground truth position of the landing pad w.r.t the drone, and repeat this process to create a dataset. The collected dataset contains 20000 images and is split 80-20 between train and evaluation sets.

**Landing pad pose estimator.** We train a model that maps top view images of a scene with a landing pad, to the relative 2D location of the landing pad w.r.t the drone in the camera frame. We use a ResNet-18 pretrained on ImageNet as the backbone for the pose regressor, and we replace the last classification layer with a regression layer outputting the $(x, y)$ relative location of the pad w.r.t drone. The model is trained end-to-end by minimizing the mean squared error (MSE) loss between the predicted location and the ground truth location. The ground truth is collected along with the images using the AirSim City simulation environment as describe before.

We train the model for 10 epochs using SGD with a fixed learning rate of 0.001, a batch size of 512, a weight decay of 1e-4, and with MSE as the objective function. The model converges fairly quickly (within the first few epochs).

**Drone Landing.** To use the pose estimator's predictions and send appropriate actions, we utilize the Multirotor API of AirSim. This allows us to control the drone by setting the desired velocity commands along all the axes (translational/rotational). Given the position of the landing pad in the scene relative to that of the drone( as output by the pose regressor) we execute the landing operation by sending appropriate velocity commands to the drone.

To generate the right velocity commands, given the relative position of the landing pad, we use a standard PID controller that computes corrective velocity values until the position of the drone matches that of the landing pad. For a pose output by the regressor treated as the setpoint $P_{set}$, and current drone pose $P_{curr}$ and at any point at time $t$, the appropriate velocity command $v(t)$ can simply be computed as follows:

$$v(t) = K_p e(t) + K_d \frac{d}{dt} e(t) + K_i * \int_0^t e(t) dt$$

where $e(t) = P_{set} - P_{curr}$, $K_p$, $K_d$, and $K_i$ are the hyperparameters of the PID controller and are manually tuned. We find that $K_p = K_d = 5$ and $K_i = 0$ to be reasonable for our task.

For realistic perturbations to the scene, similar to the 3D boosters classification experiment, we continue making use of the weather API to generate weather conditions in AirSim. This results in a variation of factors such as amount of dust or fog in the scene, allowing us to evaluate the performance of landing under various realistic conditions.

# C   Experimental Setup

## C.1   Pretrained vision models we evaluate

Here we present details of the different vision models we use in our paper. For more details on all of these, please check the README of our code at `https://github.com/microsoft/unadversarial`.

**Corruption benchmark experiments:** We use pretrained ResNet-18 and ResNet-50 (both standard and $\ell_2$-robust with $\varepsilon = 3$) architectures from [SIE+20]: `https://github.com/microsoft/robust-models-transfer`. **3D object classification in AirSim:** We use an ImageNet pretrained ResNet-18 architecture from the PyTorch's Torchvision[4] to classify all the boosted and non-boosted versions of the jets, cars, ships etc in AirSim.

**Drone landing experiment in AirSim:** We finetune an ImageNet pretrained ResNet-18 model on the regression task of drone landing. The last layer of the pretrained model is replaced with a 2D linear layer estimating the relative pad location w.r.t the drone. We collect a 20k sample dataset for training the pad pose estimation in AirSim with an $80 - 20$ train-val spilt. We use a learning rate of $0.001$, a batch size of $512$, a weight decay of $1e - 4$. We train for $10$ epochs.

**Physical world unadversarial examples experiment:** Similar to the 3D object classification experiment in AirSim, we use an ImageNet pretrained ResNet-18 architecture from Torchvision.

## C.2   Unadversarial patch/texture training details

**Patches training details** We fix the training procedure for all of the 2D patches we optimize in our paper. We train all the patches starting from random initialization with batch size of $512$, momentum of $0.9$, and weight decay of $1e-4$. We train all the patches for 30 epochs (which is more than enough as we observe that for both ImageNet and CIFAR-10, the patch converges within the first 10 epochs) with a learning rate of $0.1$ We sweep over three learning rates $\in \{0.1, 0.01, 0.001\}$ but we find that all of these obtain very similar results. So we stick with a learning rate of $0.1$ for all of our experiments..

For the classification tasks (i.e., everything but drone landing) we use the standard cross-entropy loss. For the drone landing task (landing pad pose estimation), we use the standard mean squared error loss.

**Texture training details** We now outline the process for constructing adversarial textures. We implemented a custom PyTorch module with a distinct forward and backward pass; on the forward pass (i.e., during evaluation), the module takes as input an ImageNet image, and a 200px by 200px texture; using the Python bindings for Mitsuba [NVZ+19] 3D renderer, the module returns a rendering of the desired 3D object, overlaid onto the given ImageNet image. On the backwards pass (i.e., when computing gradients), we use the 3D model's UV map[5]—a linear transformation from $(x, y)$ locations on the texture to $(x, y)$ locations in the rendered image—to approximate gradients through the rendering process. This is the same procedure used by [AEI+18] for constructing physical adversarial examples. Note that this is a simple approximation that only accounts for the location of pixels in the rendered image (i.e., ignores the effects of lighting, warping, etc.). However,

## C.3   Details of the physical world experiment

To conduct the physical-world experiments, we used a toy racecar[6], a toy warplane[7] (both from `amazon.com`) as well as two household objects: a coffeepot and eggnog container. We then printed the unadversarial patches corresponding to classes "racer," "warplane," "coffeepot," and "eggnog" on an HP DeskJet 2700 InkJet printer, at 250% scale. We adhere the patches to the top of their respective objects with clear tape (the results are shown in Figure 9b). We choose 18 distinct poses (camera positions), and for each pose took one picture of the object with the patch attached, and one

---

[4]These models can be found here `https://pytorch.org/docs/stable/torchvision/models.html`

[5]Mitsuba provides direct access to the UV map through the `aov` integrator; see the code release for more details.

[6]`https://www.amazon.com/gp/product/B07T5X69TZ/`

[7]`https://www.amazon.com/CORPER-TOYS-Pull-Back-Aircraft-Birthday/dp/B07DB3839X/`

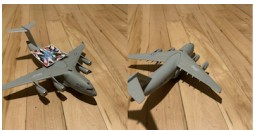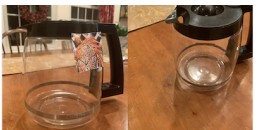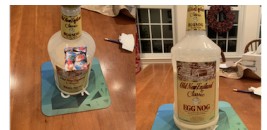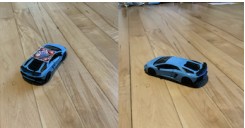

Figure 11: Photographs in different poses of the four physical objects we experimented on, with and without an unadversarial patch.

picture without (keeping the location of the patch constant throughout the experiment). Example photographs are shown in Figure 11. We evaluated a pre-trained ResNet-18 classifier on the resulting images.

## C.4 Datasets

We use two datasets across all the paper:

1. CIFAR [Kri09] https://paperswithcode.com/dataset/cifar-10.
2. ImageNet [RDS+15], with a custom (research, non-commercial) license, as found here https://paperswithcode.com/dataset/imagenet.

## C.5 Compute

We use an internal cluster containing NVIDIA 1080-TI, 2080-TI and P100 GPUs. Each experiment required no more than 1 GPU at a time.

## C.6 Replicate our results

We desired simplicity and kept reproducibility in our minds when conducting our experiments, so we use standard hyperparameters and minimize the number of tricks needed to replicate our results. Our code is available at https://github.com/microsoft/unadversarial.

# D Omitted Results

In the below figure, we show a more detailed look of the main results of the benchmarking experiments in our paper, along with useful baselines. The single color plots (e.g. the left subplot in Figure 12) report the average performance over the 5 severities of ImageNet-C and CIFAR-10-C. The multicolor bar plots (e.g. the right subplot in Figure 12) report the detail performance per severity level. The horizontal dashed lines report the performance of the pretrained models on the original (non-patched) ImageNet-C and CIFAR-10-C datasets and serve as a baseline to compare with. For both ImageNet and CIFAR as shown in Figure 13 and Figure 12, we are able to train unadversarial patches of various size that, once overlaid on the datasets, make the pretrained model consistently much more robust under all corruptions.

## D.1 Corruption benchmark main results: additional results to Figure 4b

Here we show the detailed main results for boosting ImageNet and CIFAR-10 with unadverasarial patches.

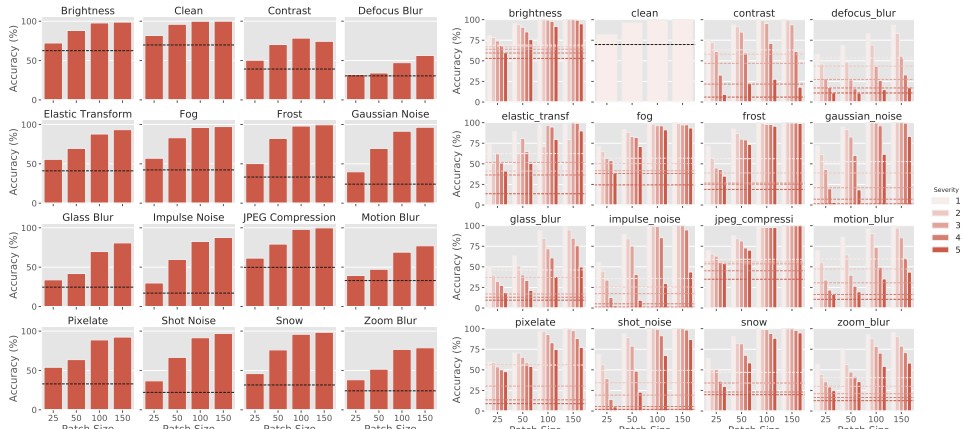

Figure 12: Robustness of a trained 2D booster over pretrained ImageNet ResNet-18 model.

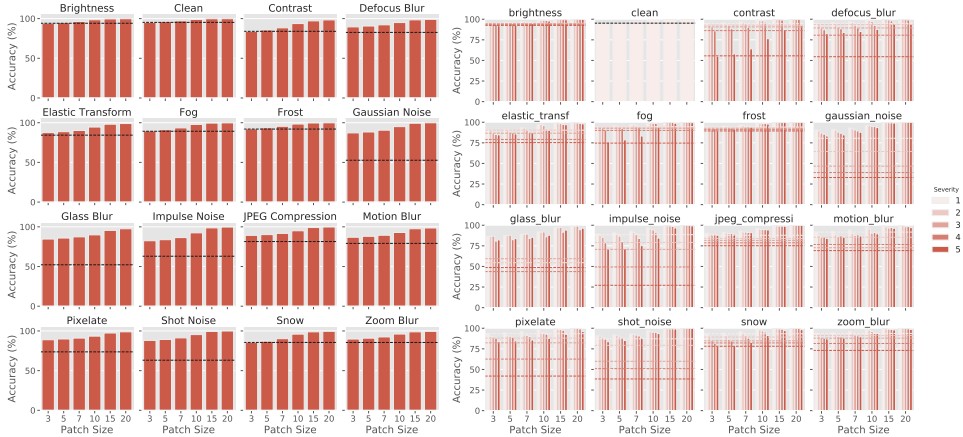

Figure 13: Robustness of a trained 2D booster over pretrained CIFAR-10 ResNet-50 model.

### D.2 Baselines

Below, we report a number of alternative ways to create patches for boosting the performance of object recognition.

### D.2.1 QR-Code

We compare our unadversarial patches to the well-known QR-Code patches. We create a QR-Code for each class of the ImageNet dataset using Python's `qrcode` package(we avoid using CIFAR-10 since the images are too small for QR-Codes to be visible and detected at all). We overlay the QR-Codes over the ImageNet validation set according in accordance to what label each image has. We add the various ImageNet-C corruption on top of the resulting images, then we use python's Pyzbar[8] package to detect the QR-Codes. The results are shown in Figure 14. The performance of QR-Codes is not comparable to what we obtain with unadversarial patches (see Figure 4b).

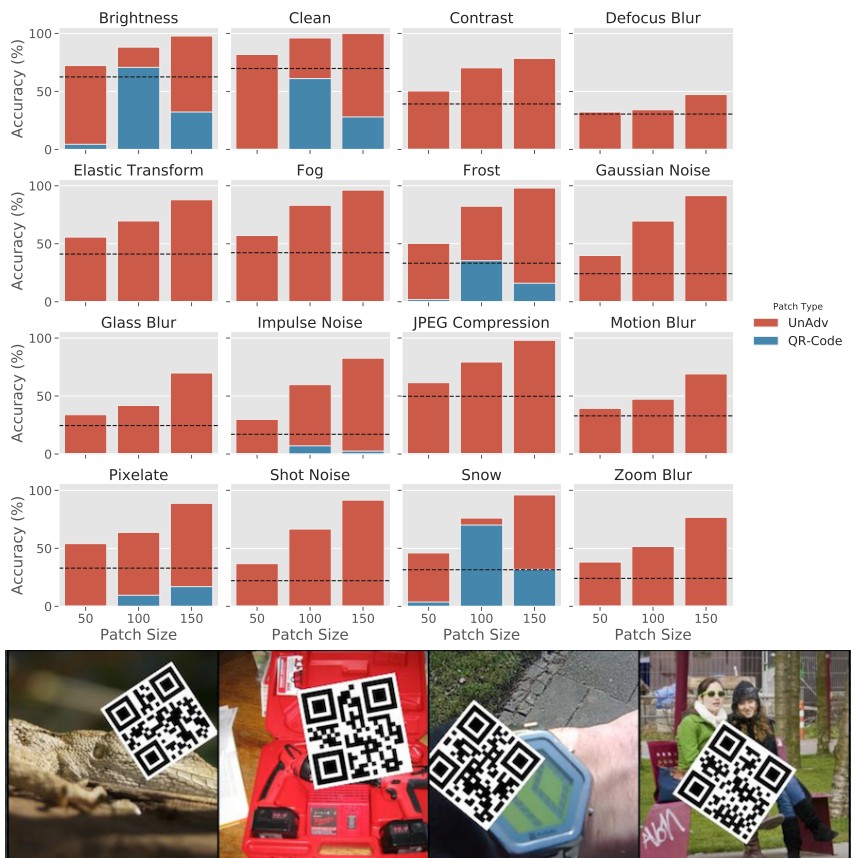

Figure 14: QR-Code boosted ImageNet results under various corruptions.

---

[8]We experiment with `OpenCV` for detecting the QR-Codes but find that `Pyzbar` leads to better performance.

### D.2.2 Best training image per class as patch

Another natural baseline that we compare with is using the best images per class in the training set of the task of interest as patches for boosting the performance of pretrained models. For example, for ImageNet classification, we simply evaluate the loss of each training image using a pretrained ImageNet model (ResNet-18 in our case), and we the image with the lowest loss per class as the patch for that class. Now we overlay these found patches onto the ImageNet validation set with random scaling, rotation, and translation (as shown in Figure 15), we add ImageNet-C corruptions, and we evaluate this new dataset using the same pretrained model we used to extract the patches. The results for ImageNet and CIFAR-10 are shown in Figure 15 and Figure 16, respectively.

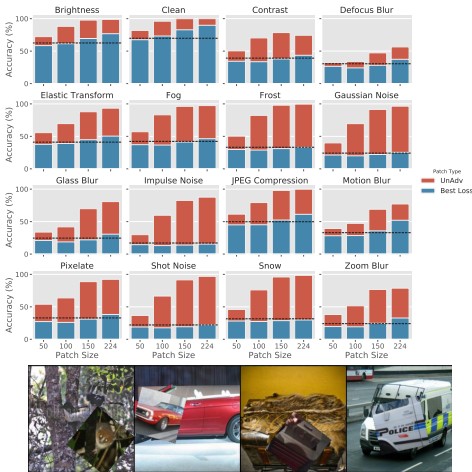

Figure 15: Best training image with translation, rotation, and scaling for ImageNet.

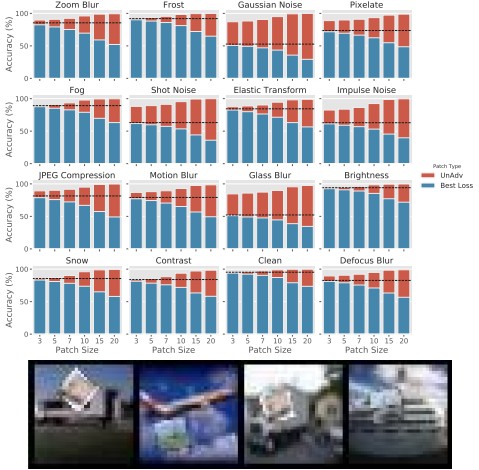

Figure 16: Best training image with translation, rotation, and scaling for CIFAR-10.

### D.2.3 Best training image vs random training image as patch

Here we investigate whether using a random image from the training set does any better than using the best-loss image as a patch. The results are shown in the below Figures. As one would expect, using a random image from the training set leads to strictly worse performance. This holds for both ImageNet and CIFAR-10 as shown in Figure 17 and Figure 18.

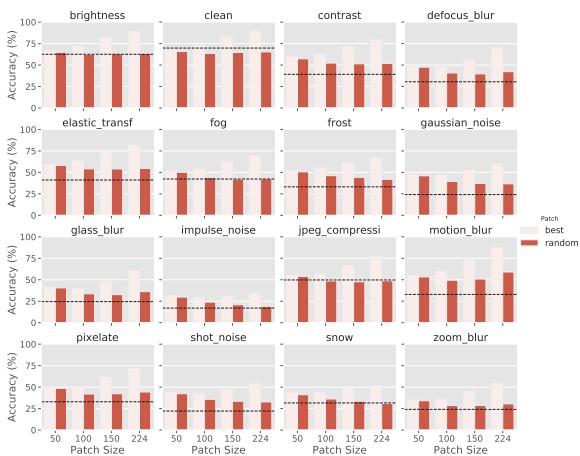

Figure 17: Best training image vs random training image with translation, rotation, and scaling.

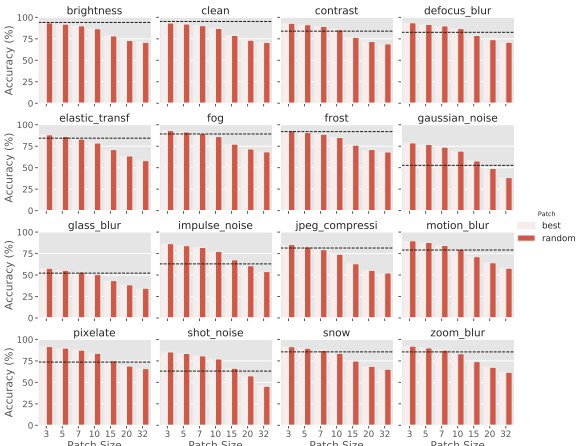

Figure 18: Best training image vs random training image with translation, rotation, and scaling.

### D.2.4   Predefined fixed-pattern unadversarial patches

This baselines is slightly different than the previous baselines since it allows the underlying classification model to be changed. Basically, we fix the set of patches to a predefined pattern (here a fixed random gaussian noise for each class), and we train a classifier on an undversarial/boosted dataset with these patches. The resulting models are consistently weaker on all corruptions of ImageNet-C and CFAR-10-C as shown in Figure 19 and Figure 20 compared to our trained patches the main paper in Figure 4b.

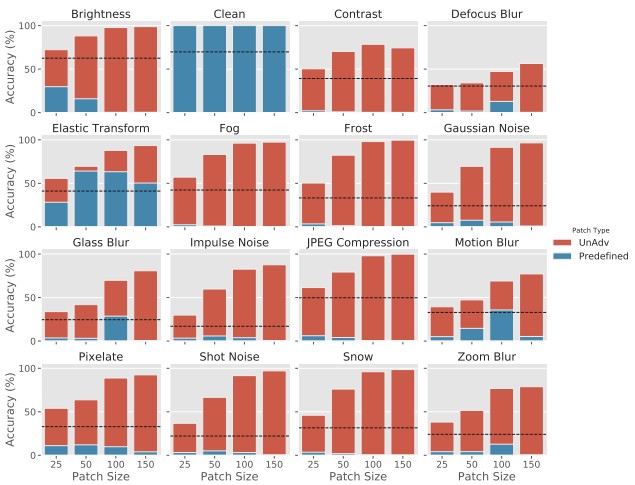

Figure 19: Robustness of an ImageNet ResNet-18 model trained on a predefined patch.

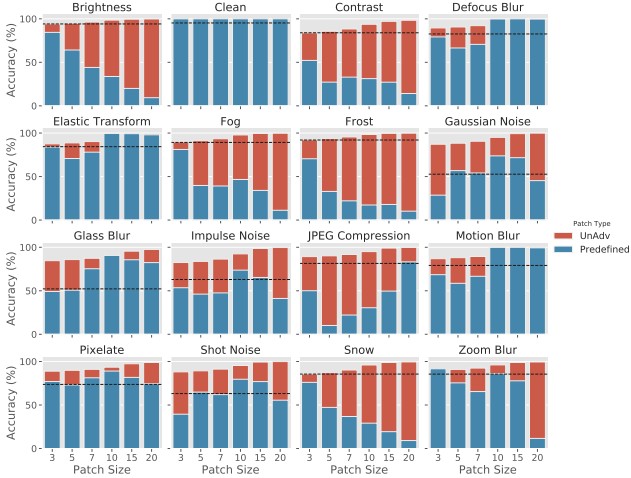

Figure 20: Robustness of a CIFAR-10 ResNet-50 model trained on a predefined patch.