# OpenReview forum: "Unadversarial Examples: Designing Objects for Robust Vision"
_NeurIPS.cc/2021/Conference — NeurIPS 2021 Poster_

### Official Review · Reviewer_np1y · 2021-07-09

**Rating:** 7
**Confidence:** 4

**Summary:**

This paper exploits the  known-sensitivity of deep models to perturbation in the input data in order to improve (rather than decrease) the performance of a pre-trained model in a classification task. Specifically, the authors propose to alter the input data (i.e. images)  by 1) adding a patch to the image 2) altering the texture of specific objects. The paper shows experimentally that this improves both in-domain performance and robustness to data corruptions.

**Limitations And Societal Impact:**

Yes

**Main Review:**

Pro:

- The idea is novel and intuitive.
- The paper is well-written, easy to follow and well-motivated.
- The experimental results shows significant boost in terms of both in-domain performance and corrupted data.

Cons.
- The technical contribution is a bit limited (mostly an extension of known techniques from the adversarial robustness literature).

Questions:
- Are the patches/texture specific to the architecture or do they generalize among different architectures and pre-trained models? Some analysis in the generalization capabilities of the unadversarial examples could make the technical contribution stronger.

- Recently, several methods have exploited the idea of "test-time" optimization to also improve both in-domain performance and out-of-domain performance. For example [1] and [2] .  While they do not act directly on the input image as here ( e.g. [1] optimizes some parameter of the models, and [2] first project the image to a latent space), the idea is somewhat similar e.g. gradient-based optimization at test time to maximize performance of the pre-trained model.  I would have loved seeing a comparison of those methods vs directly  altering the image (un-adversarial examples) since the last one could be more interpretable  but 1 and 2 does not require to directly modify the image.

[1] Tent: Fully Test-time Adaptation by Entropy Minimization ICLR 2021

[2] Semantic Segmentation with Generative Models: Semi-Supervised Learning and Strong Out-of-Domain Generalization  CVPR 2021


**Time Spent Reviewing:**

6h

---

> ### Author Response · Authors · 2021-08-10
> **Response**
>
> _Q1. Are the patches/texture specific to the architecture or do they generalize among different architectures and pre-trained models? Some analysis in the generalization capabilities of the unadversarial examples could make the technical contribution stronger._
>
> We fully agree: please see the general comment for a preliminary experiment along these lines that we believe provides further evidence of unadversarial patches’ generalization capabilities.
>
> _Q2. Recently, several methods ... require to directly modify the image._
>
> We thank the reviewer for the references and for the interesting connection to test-time adaptation which we had not made before. We will add a discussion of both the works you referenced, along with some more general literature about test-time augmentation and other similar techniques (e.g., test-time augmentation approaches, and test-time training, both of which involve changing the network during deployment in order to increase robustness).

---

> > ### Comment · Reviewer_np1y · 2021-08-17
> > **Thank you for the response.**
> >
> > Thank you for the response! I am satisfied and still believe this is a good paper.

---

### Official Review · Reviewer_mos2 · 2021-07-12

**Rating:** 7
**Confidence:** 3

**Summary:**

The paper considers a scenario where a fixed computer vision system (such as CNN based classification or regression models) is given and one tries to adapt the design of objects in order to increase their chance of being correctly detected by the system. This setting is similar to the design of marker systems for computer vision with the difference that the detection algorithm is considered fixed.

Two approaches are presented to design (i) patches that can be printed and sticked onto objects and (ii) complete textures that cover the surface of 3D objects. Both approaches follow the design of adversarial perturbations with the difference that instead of maximizing the loss of the system, it is now minimized to improve its performance instead of weakening it.

Experiments evaluate how these approaches can improve the overall performance of the given system as well as its robustness against a variety of different corruptions. Experiments are performed on synthetic 2D and 3D data, as well as on real objects, where the designed patches are printed out and attached to physical objects. Overall, they demonstrate consistent and significant improvements over non-modified objects and simple baselines.

**Limitations And Societal Impact:**

The paper discusses limitations regarding the requirement for differentiability and additional aspects that should be explored in future work, such as settings where both markers and models are optimized and the applicability to tasks other than classification and regression. It discusses impacts regarding dual use but also the potential to influence already employed models towards the better.

**Main Review:**

**Originality**

The idea to turn the table and use adversarial attacks to ones advantage is quite original and refreshing. Especially the experiments on real objects with attached stickers of the designed patches demonstrate potential use cases for such an approach. While it does not provide much methodological novelty compared to adversarial attacks, it demonstrates some novel, desirable and surprising effects of such a simple change from adversarial examples to unadversarial examples. In particular, the fact that the model still works on unmodified objects and the observation that it prefers the actual object over a conflicting patch have positive consequences for applications. Finally, it will be interesting to see how such an approach can benefit other applications besides classification and regression.

**Presentation**

The paper is well written and structured. The most important details to understand the idea are clearly presented in the main paper and the supplementary and code provide additional details which should make the results reproducible.

**Strengths**

- The paper contributes an original idea to turn adversarial attacks, which are associated with decreased robustness of CNNs, into something to increase the robustness of CNN based models (in a specific setting).
- It provides some novel and interesting results regarding conflicting signals that arises from such an unadversarial optimization of patches.
- It provides an extensive and convincing evaluation, which includes evaluation of overall performance and robustness against corruptions for varying sizes of unadversarial patches and textures, and includes an evaluation of printed patches attached to physical objects. The evaluation shows consistently positive results.
- The approach is applicable to already deployed systems to boost their performance in the presence of unadversarial markers without any negative effect in the absence of these markers.

**Weaknesses**

- It is not quite clear how realistic the scenario is for applications but this is something hard to judge objectively and I am including it only as a minor point.
- While the proposed approach has the important advantage of being applicable to pretrained, fixed models, it is still important to get a sense of how well this performs in comparison to more specialized marker approaches. The authors seem to somewhat agree with this by including QR codes as a baseline. However I do not think that this baseline accurately reflects the quality of markers for computer vision. In particular, Grinchuk, O. et al. “Learnable Visual Markers.” NeurIPS (2016) outperform QR codes and moreover, it is very close to "jointly optimizing models and unadversarial objects" (l.298) - something which according to the paper  under review should be explored in future works.

**Comments and Suggestions**

- It took me a while to get a concrete understanding of the considered setting. Until l.23 the description is rather abstract and I think it would be quite helpful to provide a concrete example already in the abstract (e.g. that which is actually considered in the paper, sticking a patch onto objects).
- l.151/Sec. 3.1 Why speak of a "threat" model here? My best guess is that this is related to adversarial attacks but in the current work I don't really see a "threat" model since there is no adversary (except maybe for image corruptions but I wouldn't call them adversarial).
- Fig. 1 and its caption provide very little context which makes it hard to understand without already knowing the considered task.
- Fig. 5 immediately left me wondering if this does not contradict results regarding adversarial patches; my concerns were well addressed in footnote 3 but I think this is a rather important point and might as well be included in the main text.
- l.217-218 a quick summary of the results that QR code detection is less robust than the proposed unadversarial examples could be included here.

**Rating**

The paper is well written, contributes an interesting idea and performs a solid evaluation thereof. The main issue that I see is a missing discussion of Grinchuk et al. 2016 and comparison to this work and other, existing marker systems besides QR codes. However, the fact that the proposed approach provides additional advantages over such existing marker systems, in particular the fact that the system gracefully falls back to its usual performance if the "marker" is missing without being affected negatively, lessens this drawback somewhat such that I recommend acceptance. Nevertheless, I would still highly recommend to at least discuss these approaches and potentially compare to them.

**Time Spent Reviewing:**

11

---

> ### Author Response · Authors · 2021-08-10
> **Response**
>
> _Q1: It is not quite clear how realistic the scenario is for applications but this is something hard to judge objectively and I am including it only as a minor point._
>
> We thank the reviewer for this comment, and hope that some of the additional experiments we have performed (in particular, the transferability one described in the general comment) provide further support for the practicality of the method.
>
> _Q2: While the proposed approach ... should be explored in future works._
>
> We are grateful for the reference provided by the reviewer and will make sure to discuss it in our upcoming revision. Learned markers are indeed similar to our proposed joint optimization setting, with the exception that we envision the joint optimization approaches to yield models that are still performant in the absence of a patch (we discuss this further below).
>
> While we agree with the reviewer that a comparison to marker-based approaches is appropriate, we chose to use QR codes since they are a universal standard that, in practice, would not need any additional implementation from a system designer, since QR code readers are readily accessible. In our opinion, comparison with highly specialized approaches that are outside of our proposed fixed-network access model are out-of-scope, since the real benefit of our method is that the system designer does not need to change anything about the classifier, which could be important for several reasons:
>
> (a) the classifier still operates as intended when the patch is not visible (as opposed to marker detection, where we have no guarantees about the performance of the classifier when the patch is out of frame)
>
> (b) The system designer may need to use a specific training algorithm due to regulatory or safety constraints, and may not be willing or able to change the training algorithm to incorporate a learned marker,
>
> (c) The fixed-design access model in combination with the black-box/transferable property of unadversarial examples means that end users who are not able to change the network can still boost its performance.
>
> _Q3. It took me a while to get a concrete understanding of the considered setting. Until l.23 the description is rather abstract and I think it would be quite helpful to provide a concrete example already in the abstract (e.g. that which is actually considered in the paper, sticking a patch onto objects)._
>
> We thank the reviewer for this valuable idea and agree with the raised concern: we will be sure to add a more concrete illustrative example to the abstract.
>
> _Q4. l.151/Sec. 3.1 Why speak of a "threat" model here? My best guess is that this is related to adversarial attacks but in the current work I don't really see a "threat" model since there is no adversary (except maybe for image corruptions but I wouldn't call them adversarial)._
>
> “Threat model” is indeed a relic from the adversarial examples literature and seems less appropriate here: we will thus change all instances of “threat model” to “access model” in our next revision, which should be more accurate.
>
> _Q5. Fig. 1 and its caption provide very little context which makes it hard to understand without already knowing the considered task. Fig. 5 immediately left me wondering if this does not contradict results regarding adversarial patches; my concerns were well addressed in footnote 3 but I think this is a rather important point and might as well be included in the main text._
>
> We will address both of these concerns in our updated manuscript, first by including an expanded caption for Figure 1 that makes sense even without context, and second by promoting footnote 3 into the main text.
>
> _Q6. l.217-218 a quick summary of the results that QR code detection is less robust than the proposed unadversarial examples could be included here._
>
> We agree and will be sure to add such a summary in our next revision.

---

> > ### Comment · Reviewer_mos2 · 2021-09-01
> > **Thank you.**
> >
> > Thank you for your response, incorporation of feedback and an additional interesting experiment on transferability. It solidifies my original assessment that this is a good paper and should be accepted.

---

### Official Review · Reviewer_eMCH · 2021-07-13

**Rating:** 9
**Confidence:** 3

**Summary:**

In this work, the authors leverage techniques from the adversarial examples literature to design patches and textures that aid in classifying an object *correctly.* In a fixed-model setting, they construct patches / textures using gradient-based methods and demonstrate effectiveness on clean CIFAR-10 and ImageNet and various robustness benchmarks such as ImageNet-C (common perturbations). The authors also compare their patches / textures to simple baselines such as QR codes, smaller-sized images from the training set, and predefined fixed patterns.

**Limitations And Societal Impact:**

Limitations and societal impact are addressed adequately in section 5.

**Main Review:**

Originality: Applying gradient-based adversarial perturbations in this setting (aiding the end user who benefits from classification) is unique and interesting. The work is an interesting application of the "adversarial patch" concept, "adversarial reprogramming," and PGD. Related work is adequately cited.

Quality: The experiments are well-designed, and the claims are well-supported by the results. The baselines presented in the appendix are thorough. The authors do a good job acknowledging practical constraints by users (i.e. line 101, recognizing that users have only limited control over physical objects). The methods are sound and well-thought out -- for example, the corruptions are applied correctly (after the patch), and the setting where patch & class conflict is studied. Code is attached to the submission and clearly organized at a glance.

Clarity: The submission is very clearly written and organized. The introduction and motivations are explained thoroughly and well. There are a few nits:

* line 82: recent work (ex: Ford and Gilmer et al. 2019) describes how adversarial examples do not have to be restricted to "carefully constructed perturbations." Although the authors of this paper implicitly acknowledge different kinds of distribution shifts in section 2, maybe the paper could benefit from changing the statement in line 82 to "Adversarial examples can include small, carefully constructed perturbations..."

* figure 5 typo: s/"red bards"/"red bar"

Finally, it would be very interesting to see how the authors ended up at the patch size they chose. Essentially, how does figure 5 change as a result of the patch size?

Significance: This work motivates a new line of work into constructing "helper" perturbations to aid in classification. The results are compelling, and the open-sourced code hopefully will make it easy for others to build on this work.

**Time Spent Reviewing:**

2 hours

---

> ### Author Response · Authors · 2021-08-10
> **Response**
>
> _Q1. line 82: recent work (ex: Ford and Gilmer et al. 2019) describes how adversarial examples do not have to be restricted to "carefully constructed perturbations." Although the authors of this paper implicitly acknowledge different kinds of distribution shifts in section 2, maybe the paper could benefit from changing the statement in line 82 to "Adversarial examples can include small, carefully constructed perturbations..."_
>
> We thank the reviewer for this comment, we agree and will make the corresponding fix in our revised manuscript.
>
> _Q2. Finally, it would be very interesting to see how the authors ended up at the patch size they chose. Essentially, how does figure 5 change as a result of the patch size?_
>
> We chose (in advance) an intermediately-sized patch for Figure 5 (10px by 10px, about 10% of the image area and the 3rd-largest patch size we studied in the paper) as it is large enough to greatly boost classification accuracy. In our next revision we can provide a version of Figure 5 for each patch size studied---we expect the result of the Figure to hold up even better (i.e., even more agreement with the true label) for all of the smaller patch sizes we tried (3x3, 5x5, 7x7, which all also boosted classification accuracy), and to degrade gradually as the patch takes over the whole image (i.e., for a 32x32 patch we would expect the classifier to uniformly agree with the patch rather than the [hidden] object).

---

### Official Review · Reviewer_5UTM · 2021-07-16

**Rating:** 6
**Confidence:** 4

**Summary:**

This work proposes and studies two techniques to generate unadversarial images (images which cause a computer vision deep learning model to more reliably give accurate predictions). One technique is for creating unadversarial patches which can be overlaid on testing images to increase the robustness of a pre-trained classifier, the other is for creating unadversarial textures for 3D meshes which when applied to 3D meshes. The work provides strong empirical evidence for unadversarial patches on ImageNet and CIFAR, including results where images are corrupted after applying the patches. Evidence is provided for unadversarial textures on simulated data and a small scale real world experiment.

**Limitations And Societal Impact:**

The authors have adequately addressed the limitations and potential negative societal impact of their work

**Main Review:**

Strengths
* The paper is very clearly written, with good figures and relevant additional details provided in the appendix.
* Strong empirical evidence for the effectiveness of unadversarial patches, including for cases where strong corruptions are applied to the images. The work provides thorough analysis with respect to patch size, simple baseline patch techniques. Further, there is evidence that when a patch is applied to a different class from the intended one it overall does not cause the model to classify according to the patch and ignore the image entirely.
* The ability to ensure visual classifier robustness for pretrained classifiers is highly desirable, especially for real world applications like fiducial marker tracking. As a result I find this work very well motivated. The drone landing simulated experiment points to the real world utility of the proposed techniques.

Weaknesses
* The takeaways that can be made from the experiments in AirSim using four ImageNet classes are not entirely clear. This is because the number of 3D models used for each class for these experiments is not specified. The results presented in Figures 6 and 7 are impressive, but it’s currently difficult to understand their significance without knowing the scale of the experiment.
* A similar weakness regarding the scale holds for the real world experiments. While it’s good to see that the patterns of improving accuracy with adversarial patches hold, experiments with only four objects do not represent strong evidence.
* While the ImageNet experiments validate the approach, there’s a lack of sufficient evidence for how applicable the proposed techniques are on real data from real world tasks. An important direct application is fiducial marker detection, and an experiment where a CNN-based fiducial marker detector is made more robust as a result of applying unadversarial patch techniques is well within scope. Such an empirical investigation would significantly strengthen this work.

Minor
Typo in figure 5 where there’s “bards” rather than “bars”

**Update After Discussion Period**

Thank you to the authors for engaging in thoughtful discussions with the reviewers. The issues pertaining to the scale of the simulation and real-world experiments remain. Making physical patches and applying them to objects (for the real world), or applying them as textures (in 3D simulation) directly shows the utility of unadversarial patches. The evidence of experiments in these settings would be more significant if there are more instances per category at test time, given that the problem is object classification and not instance classification. I therefore keep my initial rating: 6 - marginally above acceptance threshold.


**Time Spent Reviewing:**

6

---

> ### Author Response · Authors · 2021-08-10
> **Response**
>
> _Q1. Scale concerns:_
>
> We will clarify the experiments further (we use a single 3D model per class), and thank the reviewer for pointing out this ambiguity. While we agree more real-world and simulated examples would further support the practicality of unadversarial examples, each attack is expensive and hence the smaller scale of the physical-world and AirSim-simulated experiments. We are happy to study more objects in the next revision, but due to the care required in making physical experiments rigorous, the order of magnitude of the evaluation is unlikely to change. Indeed, likely for similar time/resource constraint reasons, other works that propose physical-world adversarial attacks [1, 2, 3] evaluate on the same number of (or fewer) physical-world objects.
>
> [1] Athalye et al, Synthesizing Robust Adversarial Examples
> [2] Eykholt et al, Robust Physical-world Attacks on Deep Learning Visual Classification
> [3] Sharif et al, Accessorize to a crime: real and stealthy attacks on state-of-the-art face recognition
>
> We also believe that the effectiveness and practicality of our approach are supported by the breadth of experiments. We have evaluated our proposed technique on a diverse group of settings (corrupted and standard) ImageNet and CIFAR10, multiple simulated/rendered settings in AirSim (a drone landing simulator and 3D renderings), four physical world objects, and (preliminary experiments on) ADE20K, a semantic segmentation dataset (see the response to reviewer aFae).
>
> _Q2. Applying unadversarial examples to fiducial markers:_
>
> We would like to study fiducial marker detection in the future and agree that it is an important application for future work. However, we do not believe that it is in the scope of this current work, since properly evaluating in each new setting takes considerable time, computation, and domain-specific engineering effort. Further, as mentioned above, we believe that our diverse evaluation settings already show the broad effectiveness of unadversarial examples.

---

### Official Review · Reviewer_aFae · 2021-07-17

**Rating:** 7
**Confidence:** 3

**Summary:**

This work introduces and studies "unadversarial" input modifications (patches or object textures) designed to *increase* classifier accuracy, particularly in corrupted images. The work is inspired by research on adversarial examples (in particular "adversarial patches" which trick a classifier into predicting a particular class when the patch is present in the scene, regardless of what other class might be present).

Given access to a known, fixed, pre-trained network, a small patch (or the entire object texture) are optimized by gradient descent to increase prediction accuracy. When applied, the method is shown to be quite effective for increasing classification accuracy (and performance on a simple regression task) under a variety of simulated image corruptions (blur, fog, etc.). In addition, the patch method is shown to significantly aid classification even when printed and affixed to real physical objects.

**Ethical Concerns:**

See previous section. The authors acknowledge the ethical implications of this and related works.

**Limitations And Societal Impact:**

The primary limitation addressed by the authors is that the method requires differentiability w.r.t. the patch or texture of interest. Implicit in this the need to access to the weights of a fixed, pre-trained network that is later used (unmodified) at test time. This latter implication is not always clearly considered, particularly when hypothesizing about applications of the work, as listed in the previous section. Additional care should be taken here.

As noted above, further limitations include demonstrating few results outside of classification problems. In particular, no results are shown with multiple visible patches or where size or pose of the object are predicted.

Regarding societal impact, all methods applied in this paper were first explored in the context of creating *adversarial* examples, designed to trick a classifier. This paper leverages such methods to *improve* classification when subjects can be modified, e.g. by applying a sticker to an object. The paper mentions the potential for misuse in this type of work and also notes some potential benefits when end users have the ability to increase or decrease classifier performance even without being able to control the classifier itself.

**Main Review:**

*Originality: Are the tasks or methods new? Is the work a novel combination of well-known techniques? (This can be valuable!) Is it clear how this work differs from previous contributions? Is related work adequately cited?*

Reliance on prior works is clearly and properly acknowledged throughout the paper. The method in this paper appears to be heavily inspired by "Adversarial Patches" [BMR+18], however, in addition to the novel "unadversarial" setting, this paper treads new ground in a few areas:
* Introducing an effective way to optimize "unadversarial" object texture (using the method of [AEI+18])
* Showing robustness to a wide variety of artificial image corruptions
* Showing robustness to a variety of patch sizes and viewing angles ([BMR+18] were not particularly robust here)
* Applying the method to a regression setting (drone landing pad localization)
One setting that is absent is a "black box" network where patch training is performed on one or more known networks, but testing is performed w.r.t. a *previously unseen network*.

*Quality: Is the submission technically sound? Are claims well supported (e.g., by theoretical analysis or experimental results)? Are the methods used appropriate? Is this a complete piece of work or work in progress? Are the authors careful and honest about evaluating both the strengths and weaknesses of their work?*

The experiments are well designed and clearly described, and the authors discuss limitations/strengths/weaknesses/relation to prior work throughout the text. The claims are well supported experimentally, with two caveats:

1. *Results are limited to a classification setting:* As acknowledged by the authors, nearly all results are on image classification problems. While classification is a fair and reasonable problem setting, it is not likely to be the primary setting in which the method would find real world application. And there may be non-trivial issues with extending this work to other problems, e.g. object detection. (a) While patches may increase the *saliency* of an object, it may not help (and could even harm) prediction of bounding boxes or instance segmentations. (b) Multiple patches in the same image could conflict with each other or overwhelm other prediction signals. (The results of Fig. 5 are encouraging in that patches are shown to be effective improving classification scores without dominating the visual features of the object during classification, but further experiments are needed.)

2. *Results are limited to a white-box network/dataset setting:* Again this limitation is acknowledged and is not unreasonable (though some cited hypothetical examples, e.g. line 159, would be hard to fit in this setting). The results would be stronger with an examination of the effectiveness of patches learned on Network A when seen by Network B. Also of interest here is whether the dataset is important, e.g. can patches learned with a network trained on KITTI perform well when tested on a network trained for Cityscapes?

*Clarity: Is the submission clearly written? Is it well organized? (If not, please make constructive suggestions for improving its clarity.) Does it adequately inform the reader? (Note that a superbly written paper provides enough information for an expert reader to reproduce its results.)*

The paper is well written and organized (including supplementary), figures are polished and clear, and it seems adequately informative for reproduction.

*Significance: Are the results important? Are others (researchers or practitioners) likely to use the ideas or build on them? Does the submission address a difficult task in a better way than previous work? Does it advance the state of the art in a demonstrable way? Does it provide unique data, unique conclusions about existing data, or a unique theoretical or experimental approach?*

While there is little methodologically novel here, the task and experiments are quite interesting and raise practical questions for further research. For instance, QR tags (and similar) are quite common in robotics settings; could learning "unadversarial" patches be preferable in some settings? If these results hold for black-box network settings, what impact could this have on safety and robustness with autonomous agents? The experiments in this work are a useful start.

*Bottom line*
I find the paper to be very well written, methodologically straightforward, but experimentally good, and potentially impactful both practically and toward further research. That said, the conclusions would be much more informative with two further experimental settings (object detection and transfer from network A to network B or dataset A to dataset B), and it is difficult to comprehensively judge the importance of this work without them.

*Notes*
* Line 159: Designing road signs would likely require the "black box" network setting, so it doesn't seem to be a good example for the "fixed network" setting. Similar in line 309.
* Fig. 4, Fig. 5, Fig. 6, and Fig. 9: Does "Accuracy" refer to "Top-1 Accuracy" in each of these cases? Only Fig. 7 specifically names "Top-1 Accuracy".
* Line 181, "we construct unadversarial patches of varying size": Is there a single learned patch that is scaled to various sizes when pasted on test images?
* Fig 5: "(red bards)" should be "(red bars)"
* Supplementary, Line 732: "we fix the -a- set of patches to [a] predefined pattern"

**Time Spent Reviewing:**

3

---

> ### Author Response · Authors · 2021-08-10
> **Response**
>
> _Q1. Results are limited to a classification setting:_
>
> We agree with the reviewer that the classification setting may not be the primary target for deployment of unadversarial examples---we mainly chose it due to the wide availability of robustness benchmarks and the simplicity of the engineering toolkit. We were particularly curious about/appreciative of the reviewer’s point regarding how patches may not be useful for tasks such as segmentation due to their square shape. Although obviously very preliminary (and in our opinion, out of scope of the current paper due to the additional technology required), we have performed some follow-up work analyzing semantic segmentation, the results of which are described below:
> We considered the ADE20K dataset. For each image, we chose the largest object (judged by area of the ground-truth segmentation map) and placed a trained (for segmentation)  unadversarial patch randomly within the bounding box of the object. Across all classes, the unadversarial patch boosted segmentation accuracy by 1.5% on average. Moreover, this average is skewed downwards by abnormally shaped objects (e.g., railing) where the largest inscribed square is barely visible; the unadversarial patch significantly boosts accuracy on the tree (+5% boost), plant (+12%), and sink (+3%) classes, and on perfectly rectangular objects such as rug (+24%) or wall (+10%) we see gains similar in magnitude to those in the classification setting.
>
> We hope that future work can scale unadversarial examples even further into the realms of object detection and pose estimation as well, and believe that our work sets the required foundation for these extensions.
>
> _Q2. Results are limited to a white-box network/dataset setting:_
>
> We thank the reviewer for pointing out this experiment idea, and agree that it supports the practicality of the method. Due to computational/time constraints we were unable to test transferability for all of our results, but we have verified that the “main” results indeed hold up in the black-box setting. We refer the reviewer to the general comment for details on the experiment as well as the (positive) results.
>
> We have also addressed all of the minor comments and typos pointed out by the reviewer.

---

### Author Response · Authors · 2021-08-10
**General comment**

We thank all the reviewers for their detailed and constructive comments on our paper, and have used them to improve our manuscript. We have addressed most of the reviewer questions individually as a comment, but below we summarize a few improvements/changes that are relevant to all the reviews:

**Practicality/access model/black box attacks:**

Some of the reviewers suggested that the practicality of the method would be further supported by an investigation of the transferability of unadversarial patches, i.e., whether unadversarial patches designed for a network A successfully boost accuracy on another network B (possibly of different architectures).

For the purpose of this rebuttal, we chose to study the stronger/more challenging variant of question, where A and B have different architectures: specifically, we chose A to be a ResNet-18 and B to be a ResNet-50, both trained on the ImageNet dataset. Due to computational constraints, we were unable to try every patch size and thus fixed the patch size to be 50 (i.e., less than 5% of the image).
Without implementing any additional measures to boost transferability, we observed a clean accuracy improvement of 7.4% on ImageNet, and an average robustness (ImageNet-C) improvement of 6.35%. While this is less impressive than the ResNet50 -> ResNet50 boost shown in the paper, it is a non-trivial improvement in robustness and accuracy that could further be improved by incorporating transferability-boosting techniques from adversarial robustness literature (e.g., ensembling). We will update our next revision with a full set of results across patch sizes and several other architectures.

---

### Decision · Program_Chairs · 2021-09-27

**Decision:**

Accept (Poster)

**Comment:**

Reviewers agreed that this is a solid contribution to NeurIPS. Reviewers agreed that providing more convincing evidence related to the 3D simulation and physics experiments would have significantly strengthened the paper.